# Research on China-EU equipment manufacturing Trade dependence in intra-industry specialization view

**Lingxiang Jian, Tiantian Ding***, **Wanyun Ma**

School of Maritime Economics and Management of Dalian Maritime University, Dalian, China

\* 3221171272@qq.com

## Abstract

As the foundation of the industrial economy, the equipment manufacturing industry takes an important position on the China-EU trade. Based on the analysis of the overall trend and structure of China-EU equipment manufacturing industry trade in 2007–2020, this article involves the construction of trade concentration into trade dependence metrics, and then calculate the degree of interdependence between China and EU equipment manufacturing trade in 2020. The perspective of the intra-industry specialization will be used to analyze China-EU equipment manufacturing trade dependency in 2020. The results show that: (1) Although China-EU equipment manufacturing trade has continued to grow, China had an imbalanced export structure to the EU, and electronic equipment exports are too high; (2) Regardless of import or export, the trade dependence of the EU countries on China about equipment manufacturing was higher than that of China on European countries; (3) China mainly depended on the EU about the high-end equipment manufacturing trade, which brings risks to Chinese manufacturing supply chains.

## 1 Introduction

Global division of labor and cooperation is an enduring topic and has become a major feature of the world economy. The COVID-19 pandemic and the Russian-Ukrainian War have impacted the global supply chain on the economy, and further have a complex impact on the global economy. This reality makes us once again feel that global economies are interdependent. However, after each country is embedded in the global economy, different roles have different returns in the division of labor. Differences in interests bring competition, and further may bring political conflicts. The China-US trade conflict is a typical representative event.

We selected a specific field for the relevant research. First of all, we selected China and EU countries as the economies to study. On the one hand, their economies are very large, with huge trade volume and rich trade types, which are closely linked to each other. On the other hand, we believe that as an economic union, the EU's trade policy should not only consider the whole, but also consider each country. Will different China-EU trade relations emerge because the two perspectives are not same? This is an interesting topic that attracted us very

trade flows. The only thing to note is that the HS four digit code of the equipment manufacturing products selected is between 84XX-90XX, except 8450, 8509, 8512, 8513, 8516, 8539, 8609, 8710-8715, 9003, 9004, 9006, 9008, 9010, 9020, 9021. Interested researchers can replicate the study findings in their entirety by directly obtaining the data from and following the protocol in my methods section. For a few missing data, we choose to ignore them, which has little impact on the results. The authors had no special access privileges to get the data others would not have.

**Funding:** The authors received no specific funding for this work.

**Competing interests:** The authors have declared that no competing interests exist.

much. Secondly, we selected the equipment manufacturing industry as the industry to research. The equipment manufacturing industry is a unique concept, which refers to the general term of various manufacturing industries that provide equipment for simple production and expanded reproduction in various economic departments. Since the equipment manufacturing industry determines whether other manufacturing industries can be established, it is the basic part of industry. In 2020, the equipment manufacturing trade accounted for 50.59% of the total trade between China and EU. The important reason why we study the equipment manufacturing industry is that the equipment manufacturing industry has a rich variety of products, and low technical requirements for low-end products, high technical requirements for high-end products. It is the focus of competition among big countries. In fact, the competition has gradually deepened. China put forward "Made in China 2035", trying to realize industrial upgrading in the equipment manufacturing industry such as high-end CNC machine tools, aerospace equipment, marine engineering equipment and high-tech transportation equipment. At the same time, EU countries also advocate "Reindustrialization". Among them, Germany put forward the "Industry 4.0" strategy and France put forward the "Future Industrial Plan". They focus on maintaining the position occupied by its high-precision equipment manufacturing. The cooperative and competitive relationship between China and the EU in the equipment manufacturing industry are becoming more and more complex. Studying the dependence of the equipment manufacturing industry between China and the EU can more clearly present the economic pattern of the equipment manufacturing trade between China and the EU.

Through the research on the trade dependence relationship of China EU equipment manufacturing industry, we hope to see the big from the small and explore an important question: will playing different roles in the intra industry division of labor affect the trade dependence relationship? Specifically, are low-end product producers always relying on high-end product producers as we always think? Will the production with similar prices but different characteristics be the main intra industry division of labor? And what is the difference between import dependence and export dependence? We believe that this major issue will have a direct impact on the degree of obstacles that economies face in changing their international economic roles.

Many studies on China-EU trade relations show different views. First of all, for the EU, China and the United States are the two most important trading countries. There is a long-term trade deficit in the EU's trade with China, which is very easy to cause trade frictions between China and the EU. However, Brugier believed that the EU also had a long-term trade deficit with the United States when analyzing the EU's trade strategy with China, and from the perspective of data development, the trade balance between the EU and China was more positive than the transatlantic trade balance [1]. Aiming at the trade balance between China and the EU, Jiang et al. adjusted the trade difference between China and the EU with FOB, entrepot trade and input-output tables. The results showed that the existing trade statistics methods greatly overestimated the trade surplus between China and the EU [2]. The research results of Zhou et al. showed similar conclusions. In the global industrial chain, from the perspective of added value, the EU's trade deficit with China was not large [3]. China was not only overestimated in trade surplus, but also overestimated in industrial competitiveness. As a result, Zhou et al. believed that China and the EU was highly complementary on trade. Baláž et al. put forward different views. The trade complementarity index (TCL) calculated by them could only prove that the EU and the United States have a high degree of trade complementarity, while the trade complementarity with China is not significant [4]. In recent years, China-EU trade in some specific areas had also been the focus of research. For example, Chen et al. evaluated the cost and its change trend of China-EU service trade with gravity model [5], while Covaci analyzed the development of China-EU clothing trade from 2001 to 2019 [6]. Generally

speaking, due to the more frequent economic and political conflicts in the world today, the research on the trade relationship between China and the EU has its unique significance. However, the current research mostly focused on the parts that produce conflicts, such as trade imbalance and trade competition, and there was less research on the trade dependence between China and the EU, which leaves space for the research of this article.

There are many factors that affect the trade dependence between countries. Trade dependence is an important trade relationship, which can profoundly affect economic relations and even political relations among countries. For example, the study of Hegre once again confirmed that trade interdependence reduces the risk of trade conflicts and militarization conflicts between countries [7]. Chaney found that companies have a stronger tendency to enter the nearby market when establishing the trade friction theoretical model, and trade barriers tend to increase with the increase of distance [8]. Finally, geographical distance has become an important influencing factor of trade links. Egger and Lassmann took Switzerland as an example to explore the impact of culture on trade links, and concluded that language and culture have a broad impact on trade dependence [9]. Some studies also show that more open trade policies make production more specialized, and countries become more dependent on foreign countries due to the reduction of trade diversity [10]. Neumann and Tabrizy believe that if a country increases intra industry trade, it means that there are more domestic substitutes for imported goods, which reduces import dependence [11]. This is diametrically opposed to the earlier view that the more intra industry trade, the closer the ties between countries [12]. In addition to these external factors, the trade dependence network is also vulnerable to some endogenous factors. For example, countries that are now dependent on will be more likely to be dependent on by more countries in the future. This internal trend of stronger countries is called preference dependence which was proposed by Barabasi and Albert [13]. That two countries achieve a transmission system of relations through one or more third-party countries, or form a community with sophisticated relations [14]. In order to reduce the uncertainty caused by information asymmetry, economies usually tend to form new trade dependence relationships with their partners based on the information sharing of the dependent objects, resulting in the small world phenomenon [15]. We mainly do these things in this article: first, supplement the research on China-EU trade dependence. Second, this article will divide the types of value division of intra industry trade of equipment manufacturing industry between China and EU countries according to the unit value of imports and exports of subdivided products. Third, the measurement of inter country trade dependence ignores the substitution role of other international economies. By including the global market concentration of national imports and exports into the calculation of dependence, the third-party effect can be effectively considered. Fourth, the trade network research method can visually analyze the overall trade situation, showing a complex trade structure including most countries. In view of this, this article uses the HS code 4-digit classification standard and the trade data of China-EU equipment manufacturing trade from 2007 to 2020 from UN COMTRADE database to analyze the overall scale and trade structure, calculate the mutual trade dependence of China-EU equipment manufacturing trade in 2020, establish the trade dependence relationship network under different intra industry value division to study the basic pattern of China-EU equipment manufacturing trade.

## 2 Overview of China-EU equipment manufacturing trade

### 2.1 Import and export scale of China-EU equipment manufacturing trade

In 2007, the imports and exports of China's equipment manufacturing industry to the EU was $66.519 billion and $112.098 billion respectively. In 2020, the imports and exports of China's

equipment manufacturing industry to the EU increased to $137.637 billion and $192.524 billion respectively. As shown in Fig 1, the total trade of China-EU equipment manufacturing industry generally showed a ups and downs trend. The import growth rate and export growth rate fluctuated greatly before 2012 and became more stable after 2012. This shows that China-EU equipment manufacturing trade has been more stable in recent years. From the perspective of trade balance, in 2009, the financial crisis was transmitted to the whole EU countries, resulting in a significant contraction of China's trade surplus with the EU, which was specifically reflected in the decline of 3.70% and 19.40% in imports and exports respectively. From 2010 to 2014, the trade surplus continued to decline after a brief rebound, with a decline of 77.02%, which may be affected by the post financial crisis era and the European debt crisis. From 2015 to 2020, China's trade surplus with the EU rose again. By 2020, China's trade surplus with the EU in the equipment manufacturing industry was $54.886 billion.

## 2.2 Import and export structure of China-EU equipment manufacturing trade

The equipment manufacturing industry is divided into six categories, as shown in Table 1. From the perspective of import structure, the two categories with the largest imports of China's equipment manufacturing industry to the EU are general equipment and transportation equipment. Among them, the imports of transportation equipment increased by 184.19% from 2007 to 2020, reflecting that China's demand for transportation equipment from EU countries is very large and keeps rising. In terms of export structure, electronic equipment is the largest category of China's exports to the EU. In 2007 and 2020, this category accounted for 52.56% and 45.33% of China's total exports of equipment manufacturing industry to Europe respectively, while the proportion of electrical equipment exports to Europe in the total exports of equipment manufacturing industry to Europe fluctuated the most, rising from 12.11% in 2007 to 19.15% in 2020. It reflects the rapid development of China's electrical equipment industry from 2007 to 2020. In 2007, the trade deficit of China's equipment manufacturing trade with Europe were only professional equipment trade and transportation equipment trade. In 2020, the instrument and meter trade also had a deficit. China's special equipment, transportation equipment, instrument and meter industry need to be more fully developed.

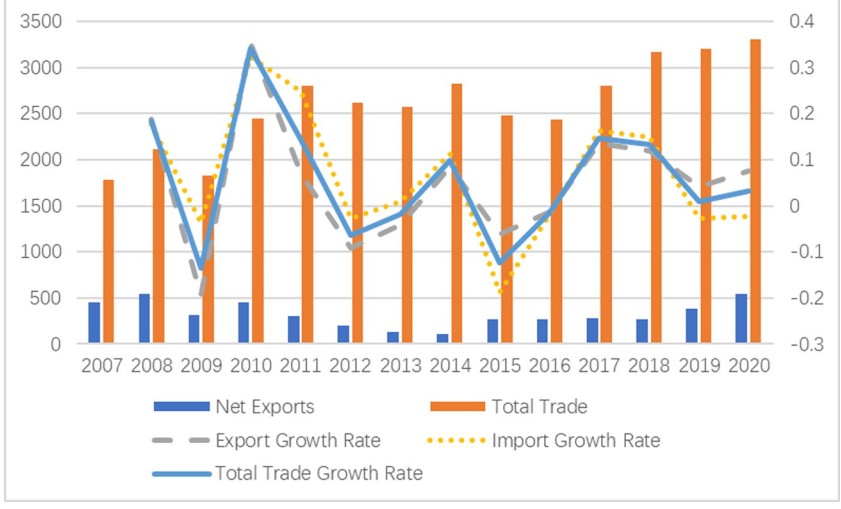

**Fig 1. Trend of China's import and export to EU equipment manufacturing industry from 2007 to 2020.**

**Table 1. China-EU import and export trade structure in equipment manufacturing industry (unit: Billion US dollars).**

| Categories | 2007 | | 2020 | |
|---|---|---|---|---|
| | Imports | Exports | Imports | Exports |
| General Equipment | 17.541 | 18.890 | 28.680 | 32.982 |
| Professional Equipment | 11.265 | 8.495 | 17.474 | 14.462 |
| Transportation Equipment | 14.789 | 7.247 | 42.028 | 11.678 |
| Electrical Equipment | 9.983 | 13.581 | 18.667 | 36.871 |
| Electronic Equipment | 8.015 | 58.916 | 14.934 | 87.264 |
| Instruments and Meters | 4.926 | 4.970 | 15.854 | 9.268 |

Overall, the trade structure of China-EU equipment manufacturing industry has not changed much from 2007 to 2020. Compared with the EU's equipment manufacturing exports to China, China's exports to the EU in the six major equipment manufacturing categories are not balanced, but this unbalanced structure slightly alleviated in 2020 compared with 2007.

## 3 Dependency relationship and intra industry trade division between China and the EU equipment manufacturing trade

Trade dependence doesn't only rely on the importance of bilateral trade volume in their respective trade volume, but also should be affected by the substitution of other countries in the international market. The market concentration measured by Herfindahl-Hirschman Index (HHI) can well measure the degree of substitution. Herfindahl Hirschman index is an index that reflects the concentration degree. It is mostly used as the concentration degree of a certain industry market and often used to measure the level of market monopoly. The calculation method is to sum the market shares of N companies in the industry in the same market squared, which can be used to calculate the concentration degree of the national import and export market as well. Compared with other concentration indexes, the Herfindahl Hirschman index has a simpler calculation method and a wider application range. At the same time, compared with the entropy index, the weight of large-scale individuals is larger, which conforms to the reality of real economy.

We establish the import and export trade dependence index *extd* and import trade dependence index *imtd* respectively. The specific calculation is as follows:

$$extd_{ij}^{k} = \frac{X_{ij}^{k}}{X_{i}^{k}} \sqrt{EC_{i}^{k}} \tag{1}$$

$$imtd_{ij}^{k} = \frac{M_{ij}^{k}}{M_{i}^{k}} \sqrt{IC_{i}^{k}} \tag{2}$$

$$EC_{i}^{k} = \sum_{j=1}^{n} \left( \frac{X_{ij}^{k}}{X_{i}^{k}} \right)^{2} \tag{3}$$

$$IC_{i}^{k} = \sum_{j=1}^{n} \left( \frac{M_{ij}^{k}}{M_{i}^{k}} \right)^{2} \tag{4}$$

The *i* and the *j* represent different countries, the *k* represents different industries, the *X* and the *M* represent exports and imports respectively, the *EC* and the *IC* represent export

concentration and import concentration respectively, and the *n* represents all countries or regions in the world.

According to the calculation results of statistical data, in 2020, the dependence rate of China's equipment manufacturing exports on the EU is 0.047, and the dependence rate of imports is 0.051, and the dependence rate of the EU's equipment manufacturing exports on China is 0.041, and the dependence rate of imports is 0.123. The dependence rate of China's equipment manufacturing exports on the EU is higher than that of the EU on China's exports, while the dependence rate of EU's equipment manufacturing imports on China is higher than that of China's imports on the EU. However, the EU is regarded as an economic whole, and the trade dependence rate of equipment manufacturing industry between China and EU member states shows a one-sided trend. The dependence rate of equipment manufacturing industry trade of EU member states on China (whether export or import) is higher than that of EU member states on China. The average export dependence rate of China's equipment manufacturing industry on EU countries is 0.002, and the average import dependence rate is 0.002. The average export dependence rate of the equipment manufacturing industry of EU countries on China is 0.035, and the average import dependence rate is 0.027.

With the gradual completion of China's industrial chain, China's equipment manufacturing industry has deeply participated in the global division of labor system. According to Greenaway et al., the division of intra industry trade can be divided into horizontal type and vertical type according to whether there are differences in the quality of import and export commodities [16]. The horizontal intra industry division refers to the production of commodities with the same quality but different types among countries, and the vertical intra industry division refers to the production of commodities with the same types but different quality among countries. Among them, the division of labor for the production of high-end products belongs to the upper vertical intra industry trade, and the division of labor for the production of low-end products belongs to the lower vertical intra industry trade. We divide the intra industry trade division according to PQV index [17], and the calculation method is as follows:

$$PQV_{ij}^{k} = \frac{PX_{ij}^{k} - PM_{ij}^{k}}{PX_{ij}^{k} + PM_{ij}^{k}} \tag{5}$$

The *PX* represents the value of the exported commodity unit, and the *PM* represents the value of imported commodity units. If $PQV_{ij}^{k} > 0.125$, country *i* and country *j* belong to upper vertical trade in the division of *K* commodity trade; if $PQV_{ij}^{k} < -0.125$, then country *i* belongs to lower vertical trade; if $-0.125 \leq PQV_{ij}^{k} \leq 0.125$, country *i* and country *j* belong to horizontal intra industry trade in *K* commodities.

According to the classification of intra industry trade in each sub industry of China-EU equipment manufacturing industry, in 2020, the equipment manufacturing industry between China and EU countries mostly carried out vertical intra industry trade, and the sub industries of horizontal intra industry trade were less than 11%. Except for six countries: Cyprus, Croatia, Estonia, Latvia, Luxembourg and Malta, the number of industries in China's equipment manufacturing trade with other EU countries in the upper vertical intra industry trade is smaller than that in the lower vertical intra industry trade. Among them, Germany, France and Italy are the EU countries that export the most types of high-end goods in the equipment manufacturing industry to China, as shown in Fig 2.

To sum up, China has a high degree of dependence on the overall equipment manufacturing trade of the EU and a low degree of dependence on the equipment manufacturing trade of EU Member States. However, most industries in China's equipment manufacturing industry carry out lower vertical intra industry trade with EU Member States.

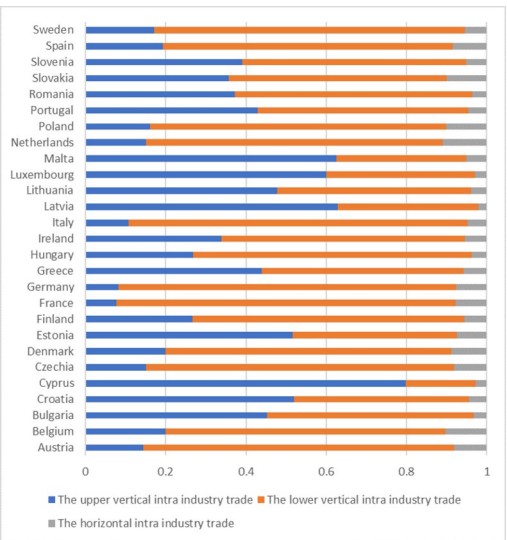

**Fig 2. Proportion of intra industry trade structure of China's equipment manufacturing industry to the EU countries in 2020.**

## 4 Network of trade dependence of equipment manufacturing industry between China and Europe

With the help of social network analysis method, we analyze the dependence of China-EU equipment manufacturing trade under different value division in 2020, explores the similarities and differences of the topology of equipment manufacturing trade network under different intra industry trade division, and further observes the interdependence between China and EU countries in different equipment manufacturing trade dependence networks.

### 4.1 Construction of trade dependence network of China EU equipment manufacturing industry

According to the types and import-export relations of intra industry trade in subdivided industries, we establish six kinds of China-EU equipment manufacturing trade dependence trade networks, which are upper vertical intra industry export dependence network, upper vertical intra industry import dependence network, horizontal intra industry export dependence network, horizontal intra industry import dependence network, lower vertical intra industry export dependence network and lower vertical intra industry import dependence network. We simplify the trade network by extracting the backbone network, and sets the standard of trade dependence rate as 0.1. If the trade dependence rate exceeds 0.1, it is considered to have significant trade dependence relation (as shown in Figs 3–8). The country names in Figs 3–8 are replaced by three letters specified by the international organization for Standardization (ISO).

### 4.2 Analysis of overall network topology

This article reflects the overall topology of six kinds of China EU equipment manufacturing industry dependence trade networks through three indicators: network density, reciprocity and average trade dependence rate (in Table 2). Network density is the ratio of actual connections to the maximum number of possible connections in the network, which reflects whether dependencies are common in the trade network. Combined with the average trade dependence of each network, it can be seen that: (1) About the intra industry trade of equipment

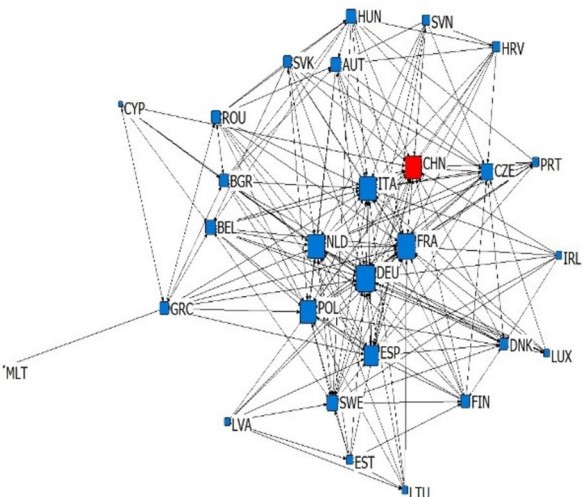

**Fig 3. Upper vertical intra industry export dependence network.**

manufacturing industry trade, China and the EU countries depends more on each other in vertical intra industry trade. The network density of horizontal import and export trade relations is 0.1858 and 0.119 respectively, and the average trade dependence is 0.082 and 0.075 respectively. Both are lower than the vertical import and export trade relationship network. (2) The dependence of China and the EU countries on the China-EU equipment manufacturing market is as follows: import high-quality products > export high-quality products > import low-quality products > export low-quality products. The high-quality and low-quality products here are relative to domestic imports and exports, which reflects that EU countries generally occupy the high-end position of the global supply chain and rely on each other's high-end products, while low-end products are more dependent on the global market. Reciprocity reflects the probability that trade related countries in the trade network have mutual dependence. According to Table 2, the reciprocity of the six trade dependence networks is low,

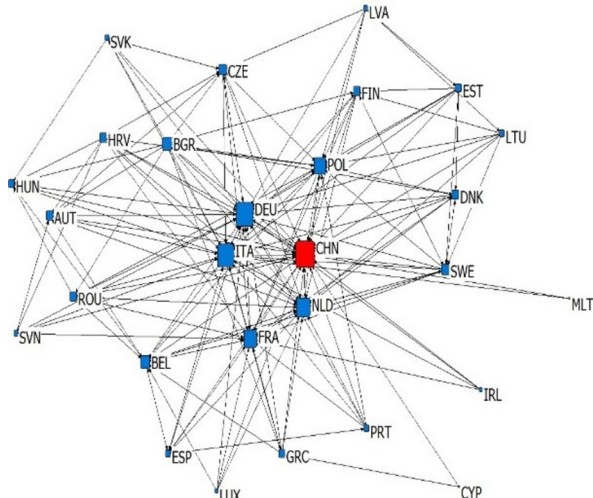

**Fig 4. Upper vertical intra industry import dependence network.**

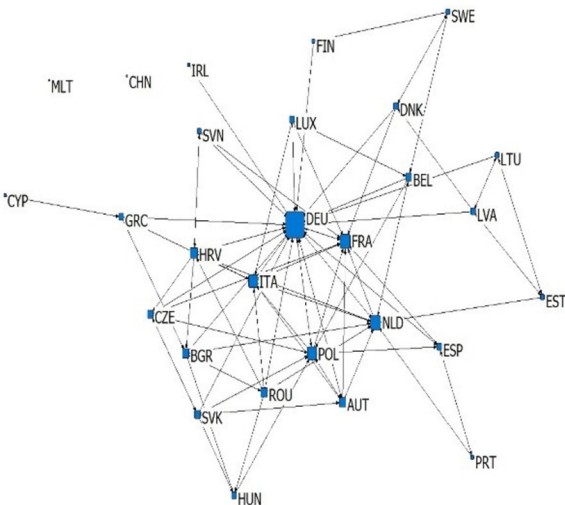

**Fig 5. Horizontal intra industry export dependence network.**

which reflects that the equipment manufacturing trade between China and the EU countries has certain scale-free characteristics.

## 4.3 Analysis of network individual centrality

Centrality reflects the status and influence of individuals in the overall trade network. In the dependence trade network, the relative out centrality refers to the dependence of a country on other countries in the network, and the relative in centrality refers to the dependence of other countries in the network on the country. According to Table 3, it can be seen that: (1) Germany, France, China and Italy are the dependent countries in the China-EU equipment manufacturing trade dependence network, while countries with low economic level, such as Bulgaria, Romania and Croatia, are the most dependent on other countries in the network. This is the common feature of the six trade dependence networks, indicating that no matter

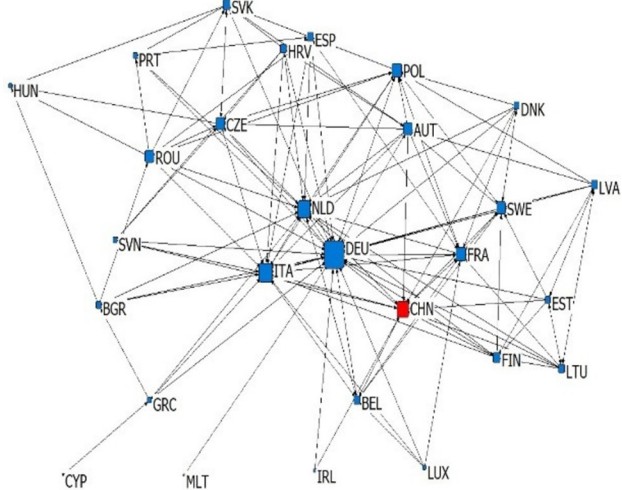

**Fig 6. Horizontal intra industry import dependence network.**

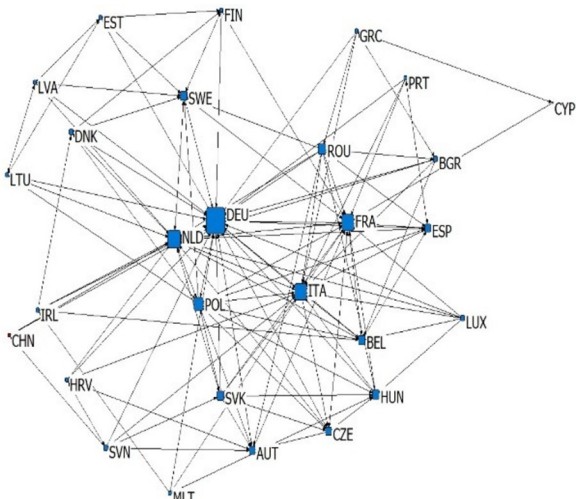

**Fig 7. Lower vertical intra industry export dependence network.**

what kind of division of value there is between countries, countries with small economies always rely heavily on foreign imports and exports in equipment manufacturing, while countries with large economies are always absolutely dependent. (2) China's dependence on the equipment manufacturing industry of EU countries is very low. The relative output centralization of four trade dependence networks, including the upper vertical export dependence network, the horizontal export dependence network, the lower vertical export dependence network and the upper vertical import dependence network, is close to 0. However, in the lower vertical import dependence network and the horizontal import dependence network, China's relative output centralization is 0.296 and 0.148, indicating that China relies on importing high-quality products and equipment manufacturing products of similar quality but different types from the EU. (3) EU countries are highly dependent on China's equipment manufacturing industry, because China's relative input centralization ranks high. Among the upper vertical import dependence networks, China's relative input

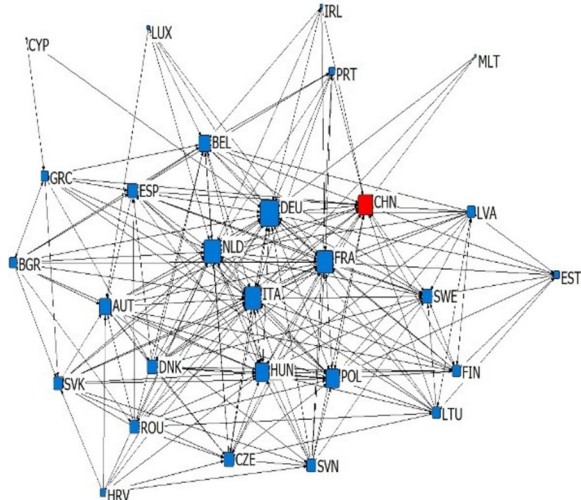

**Fig 8. Lower vertical intra industry import dependence network.**

**Table 2. Descriptive statistical characteristics of China EU equipment manufacturing trade network in 2020.**

| Networks | Network density | Reciprocity | Average trade dependence rate |
|---|---|---|---|
| Upper vertical intra industry export dependence network | 0.295 | 0.282 | 0.193 |
| Horizontal intra industry export dependence network | 0.119 | 0.286 | 0.075 |
| Lower vertical intra industry export dependence network | 0.198 | 0.293 | 0.122 |
| Upper vertical intra industry import dependence network | 0.243 | 0.227 | 0.180 |
| Horizontal intra industry import dependence network | 0.185 | 0.197 | 0.082 |
| Lower vertical intra industry import dependence network | 0.344 | 0.347 | 0.215 |

centralization is the highest, reaching 1.000, ranking first, reflecting that EU countries are significantly dependent on importing low-quality equipment manufacturing products to China. However, in the lower vertical export dependence networks and horizontal export dependence networks, China's relative input centralization is 0.000 and 0.037. EU countries have very low dependence on the export of low-quality equipment manufacturing products and equipment manufacturing products of similar quality but different types to China. Generally speaking, there is a division of value between China and EU countries in the equipment manufacturing industry production. From the perspective of trade dependence, EU countries occupy the high-end position of the supply chain to some extent, while China occupies more low-end position of the supply chain. China is bound to compete with EU countries on the road of technological catch-up. At the same time, EU countries are more dependent on China's equipment manufacturing industry than China's trade dependence on EU countries' equipment manufacturing industry. This will make China-EU equipment manufacturing trade more likely to maintain stable development in the future, showing a more stable trend than China-US equipment manufacturing trade.

**Table 3. Analysis on the centrality of China-EU equipment manufacturing trade network in 2020.**

| Networks | Top three countries in relative output centralization | Top three countries in relative input centralization | China's relative output centralization rankings | China's relative input centralization rankings |
|---|---|---|---|---|
| Upper vertical intra industry trade export dependence network | Czechia (0.407) | Germany (0.889) | 27 (0.037) | 4 (0.741) |
| | Finland (0.407) | France (0.815) | | |
| | Bulgaria (0.370) | Italy (0.778) | | |
| Horizontal intra industry trade export dependence network | Croatia (0.222) | Germany (0.740) | 28 (0.000) | 28 (0.000) |
| | Poland (0.222) | France (0.370) | | |
| | Romania (0.185) | Italy (0.296) | | |
| Lower vertical intra industry trade export dependence network | Poland (0.481) | Germany (0.741) | 28 (0.074) | 21 (0.037) |
| | Romania (0.333) | Italy (0.593) | | |
| | Slovakia (0.296) | France (0.259) | | |
| Upper vertical intra industry trade import dependence network | Bulgaria (0.481) | China (1.000) | 26 (0.074) | 1 (1.000) |
| | Germany (0.407) | Germany (0.889) | | |
| | Croatia (0.370) | Italy (0.852) | | |
| Horizontal intra industry trade import dependence network | Romania (0.333) | Germany (0.926) | 25 (0.148) | 4 (0.407) |
| | Italy (0.296) | Italy (0.630) | | |
| | Poland (0.296) | Netherlands (0.519) | | |
| Lower vertical intra industry trade import dependence network | Italy (0.519) | Germany (1.000) | 17 (0.296) | 6 (0.704) |
| | Romania (0.481) | Italy (0.852) | | |
| | Slovenia (0.444) | Netherlands (0.778) | | |

## 5 Conclusions and implications for China

Based on the analysis of the overall scale and change trend of equipment manufacturing trade between China and the EU from 2007 to 2020, and considering the substitution relationship of the global market, we calculate the interdependence between China and EU countries in the division of labor in different industries in 2020, and uses the trade network analysis method to analyze the trade dependence of equipment manufacturing industry between China and the EU from the overall level. It provides an empirical basis and useful thinking for promoting the development of China's equipment manufacturing trade with the EU. The conclusions are as follows: first, as a long-term economic partner of China's equipment manufacturing trade, the equipment manufacturing trade between the EU and China has shown a stable upward trend in the long term. The total import and export volume rose from $178.6 billion in 2007 to US $330.2 billion in 2020. And the trade structure has changed steadily, specifically, the main changes were that the value of electrical machinery exported from China to the EU increased from $13.6 billion to $36.9 billion, and the value of instruments and meters exported from the EU to China increased from $4.9 billion to $15.9 billion. China-EU equipment manufacturing trade will occupy an important position in China's equipment manufacturing foreign trade for a long time. Second, the export trade structure of China's equipment manufacturing industry to Europe is unbalanced. The proportion of electronic equipment manufacturing exports is too heavy. The export volume accounts for 45% of the total export volume of equipment manufacturing industry to Europe. On the contrary, the EU's equipment manufacturing export to China is well-distributed. In this case, China is easier to form an "export-oriented economy" that means an unhealthy industrial structure for China. Third, the dependence of China's equipment manufacturing export trade on the EU as a whole is higher than that of the EU as a whole on China. The dependence rate was 0.047 and 0.041. And the dependence of the equipment manufacturing import and export trade of EU member states on China is higher than that of China on EU member states. The sum of the average import and export dependence rates are 0.062 and 0.004. Fourth, in the division of intra industry trade, China relies more on exporting low-quality products to and importing high-quality products from EU member states, while EU Member States rely on importing low-quality products from and exporting high-quality products to China. This is manifested in two aspects: first, when EU countries conduct intra industry trade with China, there are as many as 21 countries with more products belonging to the upper vertical trade; second, among the six trade networks, China is the most dependent on the lower vertical import trade network, with a relative outward centrality of 0.296; at the same time, it is the most dependent on the upper vertical import trade network, with a relative inward centrality of 1.

Richard Emerson's power dependence theory points out that the power difference comes from the interdependence formed by individuals in the process of exchanging valuable resources. That is to say, the dependence relationship determines the individuals' influence and degree of being affected. We can see from our research conclusion that although there is a certain division of value between China and the EU, China has been relied upon strongly whether imports or exports, both on high-end products and low-end products, which is most likely due to its large market scale and production capacity. It is too absolute to say that low-end product producers are more dependent on high-end product producers. If other factors make this conclusion untenable, then in the process of industrial upgrading, low-end producers may rely on relationships to help them walk more easily.

Finally, our article still has many shortcomings. In particular, the following two points can be improved in future research. First, in some cases, some countries only undertake the final assembly work of equipment manufacturing, and their export products only appear to be of

high value on the surface, but in fact, this country does not master the core technology and does not gain major profits. Therefore, future research can progress in exploring the trade dependence between economies after calculating the value chain of the input-output table. Secondly, the importance of different products to the economy is different. A country's foreign trade dependence should not only consider the quantity of trade commodities between countries, but also consider the importance of this commodity to the economy, such as whether the replacement time of domestic production is a necessity, and whether the quantity is small but crucial in the economic structure. Therefore, it is not very appropriate to regard the importance of various products as the same status, or to completely equate their importance with their value. Future researchers need to establish a more accurate evaluation system.

## Author Contributions

**Conceptualization:** Lingxiang Jian.

**Data curation:** Tiantian Ding.

**Formal analysis:** Lingxiang Jian, Tiantian Ding.

**Funding acquisition:** Lingxiang Jian.

**Investigation:** Lingxiang Jian.

**Resources:** Wanyun Ma.

**Software:** Tiantian Ding, Wanyun Ma.

**Supervision:** Wanyun Ma.

**Writing – original draft:** Tiantian Ding.

**Writing – review & editing:** Lingxiang Jian.

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
