## [Decision Letter · Decision Letter 0]

28 Jul 2022

PONE-D-22-13626Research on China-EU Equipment Manufacturing Trade Dependence in Value Chain ViewPLOS ONE

Dear Dr. Ding,

Thank you for submitting your manuscript to PLOS ONE. After careful consideration, we feel that it has merit but does not fully meet PLOS ONE’s publication criteria as it currently stands. Therefore, we invite you to submit a revised version of the manuscript that addresses the points raised during the review process. Specifically, please try to address all the comments by the referee. Also, the conclusion with respect to China being on the lower end of supply chain should be supported and more clearly related to the data.

We look forward to receiving your revised manuscript.

Kind regards,

Petre Caraiani

Academic Editor

PLOS ONE

Journal Requirements:

Reviewers' comments:

Reviewer's Responses to Questions

**Comments to the Author**

1. Is the manuscript technically sound, and do the data support the conclusions?

Reviewer #1: Yes

2. Has the statistical analysis been performed appropriately and rigorously? 

Reviewer #1: Yes

3. Have the authors made all data underlying the findings in their manuscript fully available?

Reviewer #1: Yes

4. Is the manuscript presented in an intelligible fashion and written in standard English?

Reviewer #1: Yes

5. Review Comments to the Author

Reviewer #1: The paper is interesting and it could be considered to be published in Plos One.

Here are some comments

- The introduction should briefly place the study in a broad context and highlight why it is important. It should define the purpose of the work and its significance. The current state of the research field should be carefully reviewed and key publications cited. Please highlight controversial and diverging hypotheses when necessary.

- One of the indicators used by the author is Herfindahl-Hirschman Index. This indicator must be defined in the Methodology section. Also, the authors should explain why did he use this indicator and not others, like concentration ratios, HHI, Lorenz curve, Gini coefficient, Rosenbluth index, entropy index, Linda index, Horwath index, Lerner index etc.

- Authors should discuss the results and how they can be interpreted from the perspective of previous studies and of the working hypotheses. The findings and their im-plications should be discussed in the broadest context possible. Future research directions may also be highlighted.

6. PLOS authors have the option to publish the peer review history of their article (what does this mean?). If published, this will include your full peer review and any attached files.

Reviewer #1: No

---

## [Author Response · Author response to Decision Letter 0]

22 Sep 2022

Dear Editor and Reviewer：

Thank you for giving us the opportunity to revise our paper. We have made a lot of efforts to improve the paper according to your suggestions during this period.Here below is our description on revision according to the comments.

Part A （To Academic Editor）

1. We think it is wrong to describe our research from the perspective of value chain. Value chain should mean that different segments of a product are produced by different countries. What we study is the division of labor within the industry, which is an unequal concept. Therefore, we have revised the part of the value chain.

2. In the conclusion part, we added data explanation to make the conclusion less abrupt.

Part A （To Reviewer）

1. The introduction has been revised. We are no longer limited to the perspective of China, but focus on the study of the characteristics of trade relations. The purpose and significance of the study have also been supplemented

2. In sorting out the existing research, the research method of trade dependence is changed to the influencing factors of trade dependence, and the quality of the cited literature is also paid more attention to.

3. Added the definition of Herfindahl Hirschman index and its difference from other indexes.

4. The conclusion part has been modified to give more explanation and discussion. At the same time, it complements the direction of future research.

Thank academic editor and reviewer for your suggestions. We have learned a lot from them, which has provided a lot of help for our research. Although the existing modifications can not be perfect, we have gained a lot in the process of in-depth research.

Sincerely Yours,

Lingxiang Jian, Tiantian Ding, Wanyun Ma

---

## [Decision Letter · Decision Letter 1]

10 Nov 2022

Research on China-EU Equipment Manufacturing Trade Dependence in Intra-industry Specialization View

PONE-D-22-13626R1

Dear Dr. Ding,

We’re pleased to inform you that your manuscript has been judged scientifically suitable for publication and will be formally accepted for publication once it meets all outstanding technical requirements.

Kind regards,

Petre Caraiani

Academic Editor

PLOS ONE

Additional Editor Comments (optional):

Reviewers' comments:

Reviewer's Responses to Questions

**Comments to the Author**

1. If the authors have adequately addressed your comments raised in a previous round of review and you feel that this manuscript is now acceptable for publication, you may indicate that here to bypass the “Comments to the Author” section, enter your conflict of interest statement in the “Confidential to Editor” section, and submit your "Accept" recommendation.

Reviewer #1: All comments have been addressed

2. Is the manuscript technically sound, and do the data support the conclusions?

Reviewer #1: Yes

3. Has the statistical analysis been performed appropriately and rigorously? 

Reviewer #1: Yes

4. Have the authors made all data underlying the findings in their manuscript fully available?

Reviewer #1: Yes

5. Is the manuscript presented in an intelligible fashion and written in standard English?

Reviewer #1: Yes

6. Review Comments to the Author

Reviewer #1: (No Response)

7. PLOS authors have the option to publish the peer review history of their article (what does this mean?). If published, this will include your full peer review and any attached files.

Reviewer #1: No

---

## [Editor Report · Acceptance letter]

15 Nov 2022

PONE-D-22-13626R1 

Research on China-EU Equipment Manufacturing Trade Dependence in Intra-industry Specialization View 

Dear Dr. Ding:

I'm pleased to inform you that your manuscript has been deemed suitable for publication in PLOS ONE. Congratulations! Your manuscript is now with our production department. 

Kind regards, 

on behalf of

Dr. Petre Caraiani 

Academic Editor

PLOS ONE